# Branching Memory: Task-Specific Expansion for Continual Learning in Large Language Models

## Abstract

Large Language Models (LLMs) face the challenge of catastrophic forgetting in continual learning scenarios, where learning new tasks often overwrites previously acquired knowledge, leading to performance degradation and limiting their applicability in dynamic task environments. Existing approaches can be categorized into rehearsal-based, regularization-based, and architecture-based methods. Among these, architecture-based methods are more suitable for LLMs as they dynamically adjust model structures to handle large-scale parameters and task interference. However, existing methods often struggle with parameter efficiency and fail to fully leverage the Transformer architecture's characteristics. In this work, we propose Branching Memory, a novel method that leverages the organization of knowledge within transformer models. By modeling knowledge as key-value (KV) representations within the FFN layers, our approach dynamically allocates dedicated capacity for new tasks, allowing the model to store and integrate task-specific knowledge without overwriting existing information. To further improve knowledge retention and reduce task interference, we employ an orthogonality-based regularization strategy to stabilize training and minimize parameter conflicts. Experimental results on standard continual learning benchmarks demonstrate that Branching Memory achieves superior performance with enhanced parameter efficiency. On short-sequence tasks with T5-Large, Branching Memory with regularization achieves 76.6% average accuracy, outperforming baseline methods. Extended evaluations on LLaMA2-7B and 15-task long sequences validate the method's scalability and effectiveness across different model architectures and task lengths. The method's practical advantage lies in its balanced trade-off between performance, parameter efficiency, and inference simplicity in continual learning scenarios.

## 1 Introduction

Large Language Models (LLMs) have become the cornerstone of natural language processing (NLP). However, in dynamic task scenarios, they face the challenge of catastrophic forgetting McCloskey & Cohen (1989), where learning new tasks overwrites knowledge of previous ones, leading to performance degradation Wang et al. (2022a). This problem is particularly severe in longer task sequences due to the large parameter space and complex feature sharing mechanisms in LLMs Brown et al. (2020). Addressing catastrophic forgetting in LLMs is crucial to enabling their effective deployment in real-world applications, especially in dynamic and resource-constrained environments. Continual Learning (CL) Aljundi (2019) provides a promising solution by incrementally acquiring new knowledge from sequentially arriving tasks while retaining the knowledge of previous ones Wu et al. (2024). This paradigm provides a possible solution for LLMs, as it avoids full retraining on all tasks, significantly reducing computational costs, and enabling more efficient task adaptation.

Existing CL methods Wu et al. (2024) focus mainly on three approaches: rehearsal-based methods, regularization-based methods, and architecture-based methods. **Rehearsal-based methods** replay data from previous tasks during the training of new tasks to maintain knowledge of the old tasks Lopez-Paz & Ranzato (2017a). However, they require additional storage resources and raise

privacy concerns. **Regularization-based methods** impose constraints on key parameters of the model using data from previous tasks (*e.g.*, Elastic Weight Consolidation, EWC Kirkpatrick et al. (2017)). However, simple regularization struggles to achieve effective feature sharing and interference control in long task sequences. **Architecture-based methods**, such as Mixture of Experts (SEMA) Wang et al. (2025) and bert2BERT Chen et al. (2021), dynamically expand the capacity of the model to reduce the interference of tasks Houlsby et al. (2019a), effectively reducing the interference between tasks compared to the other two approaches.

However, architecture-based methods still have shortcomings in fully leveraging the potential of transformer architectures Vaswani et al. (2017) and there is room for improvement to mitigate catastrophic forgetting. Furthermore, current CL methods face significant limitations when applied to LLMs Wu et al. (2024); Zheng et al. (2025). The massive parameter scale of LLMs exacerbates catastrophic forgetting, and many existing CL methods (*e.g.*, EWC) do not adequately address complex interference between tasks. Moreover, LLMs need to share feature representations across multiple tasks, but existing architecture-based methods focus excessively on task adaptation, neglecting the balance between feature sharing and task isolation. Recently, hybrid methods that combine low-rank adaptation and regularization Wang et al. (2023a) have partially alleviated these problems, but memory retention and parameter efficiency in LLMs still require further improvement.

In this context, this paper proposes the **Branching Memory** method, which leverages the memory retention capabilities of the feed-forward networks (FFNs) Geva et al. (2020) in transformer models. We reinterpret the memory in the Transformer FFNs as a set of Key-Value pairs (KV), turning task knowledge into memory-efficient storage units embedded in the model. For continual learning, we dynamically allocate additional KV pairs per task by expanding the intermediate dimension of the FFN, providing task-specific memory without a blanket growth of total capacity. Only the active task's memory components are updated, providing additional feature storage for new tasks while reducing computational and storage overhead.

Our method achieves superior parameter efficiency, with each task adding only 196,608 parameters (0.22% of T5-Large) while maintaining competitive performance. Furthermore, we incorporate two optimization strategies: a special initialization strategy to minimize parameter overlap, and orthogonality constraints as a regularization technique to reduce interference between task branches. Extensive experiments validate our approach across multiple model architectures and task sequences.

Our contributions are summarized as follows:

- We propose the **Branching Memory** method, which introduces task-specific lightweight memory branches in Transformers to expand memory capacity and mitigate catastrophic forgetting. By incorporating **regularization** and **initialization** strategies, interference between tasks is effectively reduced.

- Our method achieves superior parameter efficiency and task isolation. With branch dimension $d'_m = 16$, each task adds only 196,608 parameters (vs 294,912 in IncLoRA), significantly reducing computational and storage costs while maintaining inference simplicity through model consolidation.

- Extensive experiments on standard continual learning benchmarks demonstrate that Branching Memory achieves competitive performance with enhanced parameter efficiency. On short-sequence tasks, regularization method achieves 76.6% average accuracy, while extended evaluations on LLaMA2-7B and 15-task sequences validate scalability and cross-architectural effectiveness.

## 2 RELATED WORK

Continual Learning Wu et al. (2024); Zheng et al. (2025) enables models to incrementally learn new tasks while retaining knowledge of prior tasks. Existing methods can be broadly categorized into three types:

**Rehearsal-based methods.** Rehearsal-based methods store and replay data or features from previous tasks to preserve knowledge. For example, GEM Lopez-Paz & Ranzato (2017b) constrains gradient directions to avoid interference, while LAMAL Bai et al. (2022) leverages language mod-

eling for task replay. Despite their effectiveness, these methods suffer from high storage overhead, privacy concerns, and difficulties in mitigating interference in long task sequences.

**Regularization-based methods.** These methods mitigate forgetting by penalizing changes to critical model parameters. For instance, EWC Kirkpatrick et al. (2017) applies regularization to important weights, while O-LoRA Wang et al. (2023a) incorporates low-rank adaptation. However, these methods struggle to balance feature sharing and interference control, especially in long task sequences.

**Architecture-based methods.** Architecture-based methods dynamically expand or modify models to reduce task interference. For example, Prefix-Tuning Li & Liang (2021), LoRA Hu et al. (2022), and Adapters Pfeiffer et al. (2020) introduce task-specific parameters, while MoE Shazeer et al. (2017) uses expert gating to isolate tasks. While effective in task adaptability and parameter efficiency, these methods face trade-offs in generalization, task isolation, and computational overhead.

## 3 BACKGROUND

This section reviews continual learning fundamentals and the Transformer architecture, focusing on the role of FFNs in memory and task representation.

### 3.1 CONTINUAL LEARNING

Continual Learning (CL) enables models to learn tasks sequentially, retaining prior knowledge while learning new ones Wu et al. (2024). For supervised CL, tasks $D(t)$ arrive sequentially, and the model only has access to the current task $D(t)$. A major challenge in CL is **catastrophic forgetting**, where learning new tasks degrades performance on earlier ones.

Two key metrics are used to evaluate CL Zheng et al. (2025):

- **Average Accuracy (AA):** Measures overall performance across tasks:

$$\text{AA}_t = \frac{1}{t} \sum_{i=1}^{t} a_{i,t}. \tag{1}$$

- **Backward Transfer (BWT):** Quantifies the impact of new task learning on previous tasks:

$$\text{BWT}_t = \frac{1}{t-1} \sum_{i=1}^{t-1} (a_{i,t} - a_{i,i}). \tag{2}$$

Higher AA indicates better overall retention, while a BWT closer to zero reflects minimal forgetting.

### 3.2 TRANSFORMER ARCHITECTURE

The Transformer Vaswani et al. (2017) is built on Multi-Head Attention (MHA) and Feed-Forward Networks (FFNs). MHA captures dependencies across input token positions, while FFNs enhance expressive power by processing features through nonlinear layers. FFNs dominate the parameter footprint in Transformers Geva et al. (2020), making them critical for feature storage.

FFNs are structured as:
$$\text{FFN}(x) = f(xW_1 + b_1)W_2 + b_2, \tag{3}$$
where $f$ is a nonlinear activation, and $W_1, W_2$ are weight matrices. The hidden dimension $d_m$ governs the FFN's feature storage capacity, which is crucial for multi-task learning.

Research Geva et al. (2020) suggests that FFNs act as a **key-value memory structure**:
$$\text{Memory}(x) = \text{softmax}(x \cdot K^T) \cdot V, \tag{4}$$
where $K$ and $V$ represent key and value matrices. While this memory structure supports task-specific mappings, its capacity is limited, leading to overwriting of old task representations in continual learning. To address this, we propose a task-branch memory expansion framework that leverages FFN's structure for efficient and isolated memory allocation.

# 4 METHOD

The FFN in a Transformer acts as a potential "memory unit" due to its high-dimensional hidden layers, which can effectively store task-specific patterns. However, directly increasing the hidden layer dimension $d_m$ to expand the FFN's memory capacity, while simple and effective, significantly increases computational and storage costs, particularly for large-scale models.

To address this issue, we propose a **key-value-based memory expansion** method. Instead of directly increasing the hidden layer dimension, we introduce task-specific learnable key-value pairs for each task $t$. These learnable components allow the model to effectively expand its memory capacity while maintaining parameter efficiency and computational economy. This approach isolates task-specific representations, reducing interference between tasks.

Suppose the original FFN is represented as:

$$\text{FFN}_{\text{old}}(x) = f(xW_{\text{old1}} + b_{\text{old1}})W_{\text{old2}} + b_{\text{old2}}, \tag{5}$$

where $W_{\text{old1}} \in \mathbb{R}^{d \times d_m}$, $W_{\text{old2}} \in \mathbb{R}^{d_m \times d}$, and $d_m$ is the hidden layer dimension.

We introduce a lightweight branch for task-specific expansion:

$$\text{FFN}_{\text{branch}}(x) = f(xW_{\text{key}}^{(t)} + b_{\text{key}}^{(t)})W_{\text{value}}^{(t)} + b_{\text{value}}^{(t)}, \tag{6}$$

where $W_{\text{key}}^{(t)} \in \mathbb{R}^{d \times d_m'}$, $W_{\text{value}}^{(t)} \in \mathbb{R}^{d_m' \times d}$, and $d_m'$ corresponds to the hidden dimension of the branch representation.

The total output of the FFN with this branch becomes:

$$\text{FFN}_{\text{total}}(x) = \text{FFN}_{\text{old}}(x) + \text{FFN}_{\text{branch}}(x). \tag{7}$$

This expansion approach using lightweight branches provides significantly higher parameter efficiency compared to directly increasing the hidden layer dimension. With $d_m' = 16$, each task adds only 196,608 parameters, representing just 0.22% of T5-Large's total parameters.

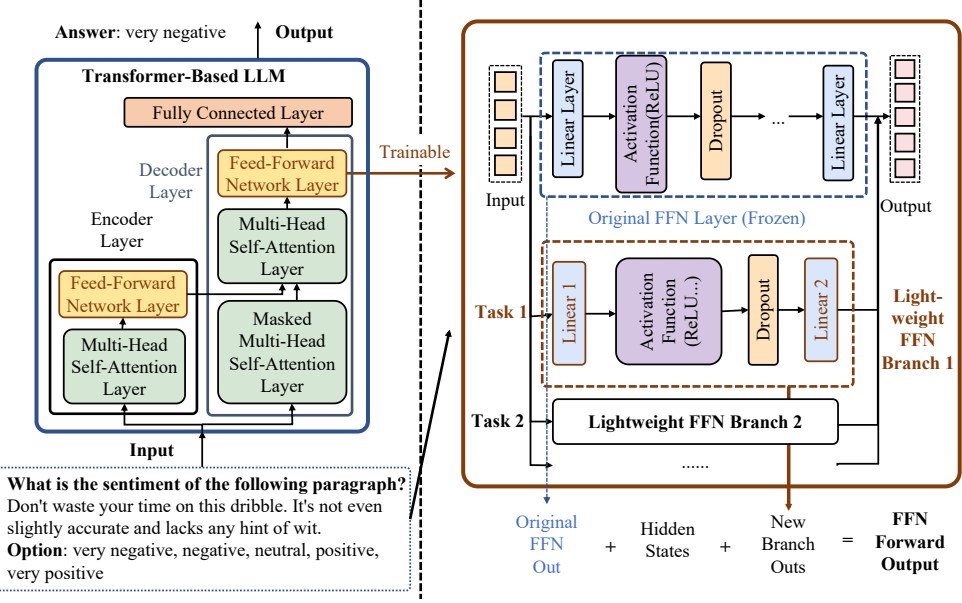

Figure 1: The framework of the Branching Memory method for continual learning in LLMs. Most parameters in the transformer are frozen, while low-rank branches are introduced in FFNs for each task. Each task corresponds to its own gradient subspace in these branches, while the parameters of other branches remain untouched.

## 4.1 Overall Framework

In the context of continual learning, the model needs to sequentially handle multiple tasks $T_1, T_2, \ldots, T_t$. To this end, our framework built upon a pretrained transformer model is designed as follows:

**Freezing pretrained Parameters and Previous Task Branches**: The pretrained transformer parameters (*e.g.*, embedding layers, multi-head self-attention modules, and feed-forward layers) as well as the lightweight memory branch parameters from all prior tasks are frozen to preserve previously learned knowledge and avoid interference.

**Introducing Task-Specific lightweight memory branches**: For the current task $T_t$, an independent lightweight memory branch is introduced in the FFN of each transformer layer. Its parameters are denoted as $(W_{1,\text{branch}}^{(t)}, b_{1,\text{branch}}^{(t)}, W_{2,\text{branch}}^{(t)}, b_{2,\text{branch}}^{(t)})$. Only the parameters of the current task branch are trained, while the original FFN and previous task branches remain unchanged.

**Output Design**: The output of the extended FFN is the sum of the frozen original FFN output and the task-specific lightweight memory branch output:

$$\text{FFN}_{\text{total}}^{(t)}(x) = \text{FFN}_{\text{original}}(x) + \text{FFN}_{\text{branch}}^{(t)}(x), \tag{8}$$

where $\text{FFN}_{\text{original}}(x)$ retains pretrained knowledge and all previous tasks' knowledge and $\text{FFN}_{\text{branch}}^{(t)}(x)$ learns task-specific patterns for the current task.

For the training of each task, the input data first passes through the first LayerNorm He et al. (2016) ($\text{LN}_1$) and the multi-head attention module (MHA), with the result added to the original input through residual connections. Subsequently, it passes through a second LayerNorm ($\text{LN}_2$) and is fed into the extended FFN. Within the extended FFN, the frozen original FFN generates $\text{FFN}_{\text{original}}(x)$, while the task-specific lightweight memory branch produces $\text{FFN}_{\text{branch}}^{(t)}(x)$. These outputs are summed and returned to the previous layer via residual connections. This design isolates task-specific parameters while leveraging residual connections to enhance model stability and data flow consistency.

Figure 1 provides an intuitive illustration of the proposed framework and its task-handling process in continual learning. By employing independent lightweight memory branches, the influence of new tasks on previous ones is minimized Rusu et al. (2016). Additionally, the parameter-efficient design Houlsby et al. (2019b) results in minimal increases in storage and computational costs for new branches. Furthermore, the specific patterns of each task are learned by their corresponding branches, thereby maximizing the retention of pretrained knowledge and representations from previous tasks, leading to strong adaptability.

## 4.2 Optimizations for lightweight memory branches

Based on the Branching Memory method and inspired by existing continual learning methods Wu et al. (2024); Zheng et al. (2025); Wang et al. (2024), we further propose two optimization strategies: the initialization strategy and the regularization strategy. These strategies aim to reduce interference between new and old task branches and improve the model's ability to retain knowledge from previous tasks. The specific structure is shown in Figure 2.

### 4.2.1 Regularization Strategy

To suppress interference between new and previous branches, we incorporate a regularization strategy to constrain the training parameters. Specifically, we introduce an orthogonality regularization Wang et al. (2023a) term to ensure that the lightweight memory branch parameters for the new task remain orthogonal or sufficiently distant from those of previous tasks, thereby reducing the interference of new tasks on the feature representations of prior tasks.

Let the lightweight memory branch parameters for the $t$-th task be $W_{\text{task}}^{(t)} \in \mathbb{R}^{r \times d}$, and let the parameters for the $i$-th previous task be $W_{\text{task}}^{(i)}$. The orthogonality regularization is defined as:

$$\mathcal{L}_{\text{ortho}}^{(t)} = \sum_{i=1}^{t-1} \left\| \left( W_{\text{task}}^{(t)} \right)^{\top} W_{\text{task}}^{(i)} \right\|_F^2, \tag{9}$$

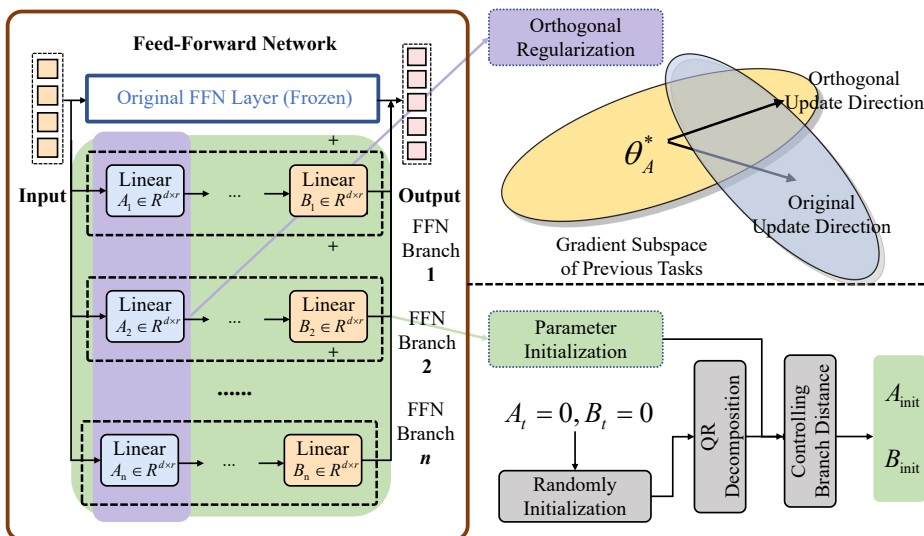

Figure 2: Regularization and initialization extensions of the Branching Memory method. Orthogonality constraints reduce interference, while distance-based initialization minimizes the gap between new and old branch parameters.

where $\|\cdot\|_F$ denotes the Frobenius norm.

Combining this with the primary task learning objective $\mathcal{L}_{\text{task}}$, the complete training objective is given by:

$$\mathcal{L} = \mathcal{L}_{\text{task}} + \lambda \mathcal{L}_{\text{ortho}}^{(t)}, \tag{10}$$

where $\lambda$ is the regularization coefficient that controls the impact of the regularization term.

### 4.2.2 INITIALIZATION STRATEGY

We explored an initialization strategy, but found it provided only marginal gains compared to Branching Memory. Detailed implementation of the initialization approach is provided in the appendix A.

## 5 EXPERIMENTS

This section presents the experimental setup and results of the proposed lightweight memory branch expansion method, including its regularization and initialization extensions. The experiments evaluate performance on both short-sequence and long-sequence continual learning tasks, demonstrating the method's effectiveness in mitigating catastrophic forgetting and enhancing memory retention. Additionally, we analyze the impact of key hyperparameters on the proposed method's performance.

### 5.1 EXPERIMENTAL SETUP

**Models and Datasets:** We evaluate our approach using the CL benchmark Zhang et al. (2015) for language models,using T5-large Deng (2019) as the backbone model, evaluating it on both short-sequence and long-sequence tasks. In Short-sequence tasks, Five text classification datasets are adopted, including AG News, Amazon Reviews, Yelp Reviews, DBpedia, and Yahoo Answers, as proposed by Zhang et al Zhang et al. (2015), following LFPT5 Qin & Joty (2021), and evaluates it across three different learning orders. The specific task sequences, datasets, and experimental parameters are detailed in appendix A. Following the method of Razdaibiedina *et al.* Wang et al. (2023b), 1000 samples were randomly selected for training for each task, and 500 samples per class were retained for validation. In Long-sequence tasks, 15 datasets Razdaibiedina et al. (2023) are selected from GLUE Wang et al. (2018), SuperGLUE Wang et al. (2019), and other benchmarks

such as IMDB and BoolQA Maas et al. (2011) (details provided in appendix A). Furthermore, we validate the generalizability of the method in Llama-7B.

**Evaluation Metrics:** Average accuracy (AA) and backward transfer (BWT) are used to evaluate the performance of the models.

**Baseline Methods:** We compare the proposed method with the following baselines (detailed descriptions are provided in the appendix A): MTL, SeqFT de Masson D'Autume et al. (2019), SeqLoRA, IncLoRA, Replay, EWC Kirkpatrick et al. (2017), LFPT5 Qin & Joty (2021) , L2P Wang et al. (2022b), O-LoRA Wang et al. (2023a), InfLoRA Liang & Li (2024) and GainLoRA Liang & Li (2025). Details of the baselines are provided in the appendix A.

**Variants of the Proposed Method:** We evaluate the following variants of the proposed method: The basic **Branching Memory** method, **Branching Memory (Initialization):** method utilizing a distance-sensitive initialization strategy for new branches, and **Branching Memory (Regularization):** method that adds orthogonality constraints between lightweight memory branches corresponding to different tasks All experiments report the average results from three runs with different random seeds.

**Statistical Rigor:** All experiments were conducted with 3 independent runs using different random seeds. Statistical significance was confirmed via t-test ($p < 0.05$) for all reported improvements over baselines.

## 5.2 Experimental Results

### 5.2.1 Results on Short-Sequence Tasks

Table 4 summarizes the average accuracy of the proposed methods and baselines on the short-sequence benchmark. Our Branching Memory method will be referred to as BM in subsequent tables.

From the results in the table, it can be observed that Branching Memory significantly improves training accuracy compared to most baselines in short-sequence tasks, with an average increase in accuracy of over 20% compared to methods such as IncLoRA and Replay. Although the basic Branching Memory method does not outperform the O-LoRA method combined with regularization, our approach shows further improvement when both regularization and initialization are applied, with the regularization method achieving the highest AA, approaching multi-task learning performance. This indicates that the regularization and initialization strategies play a crucial role in enhancing model performance.

Table 1: Performance on short-sequence continual learning benchmark with T5-large

| Method | Order-1 | Order-2 | Order-3 | Avg |
|---|---|---|---|---|
| MTL | 80.0 | 80.0 | 80.0 | 80.0 |
| SeqFT | 18.9 | 24.9 | 41.7 | 28.5 |
| SeqLoRA | 44.6 | 32.7 | 53.7 | 43.7 |
| IncLoRA | 68.0 | 65.8 | 72.8 | 68.9 |
| Replay | 55.2 | 56.9 | 61.3 | 57.8 |
| EWC | 48.7 | 47.7 | 54.5 | 50.3 |
| LFPT5 | 67.6 | 72.6 | 77.9 | 72.7 |
| L2P | 60.3 | 61.7 | 61.1 | 60.7 |
| O-LoRA | 75.4 | 75.7 | 76.3 | 75.8 |
| **InfLoRA** | **77.5** | 78.2 | 77.8 | 77.8 |
| **GainLoRA** | **79.5** | 79.1 | 79.6 | 79.4 |
| **BM** | 74.9 | 74.0 | 74.1 | 74.3 |
| **BM (Init.)** | 75.2 | 74.3 | 74.7 | 74.7 |
| **BM (Reg.)** | **77.3** | **77.1** | **75.5** | **76.6** |

### 5.2.2 Backward Transfer Analysis

Table 2 compares the BWT of different methods. Results show that the Branching Memory method significantly improves memory retention compared to existing methods, with the regularization extension achieving near-zero BWT values. This indicates that both the Branching Memory method and orthogonality-based regularization can enhance the model's memory retention in continual learning.

### 5.2.3 Results on Long-Sequence Tasks

To comprehensively evaluate scalability, we conduct experiments on two extended task sequences: **Long-Order-1** (15 tasks) and **Long-Order-2** (15 tasks) with T5-large model. Figure 3 shows the learning trajectories across both sequences.

Table 2: Backward transfer (BWT) on Performance on short-sequence continual learning benchmark with T5-large.

| Method | Order-1 | Order-2 | Order-3 | Avg |
|---|---|---|---|---|
| IncLoRA | -13.2 | -17.7 | -8.8 | -13.2 |
| Replay | -10.7 | -11.1 | -10.9 | -10.9 |
| **BM** | -5.2 | -6.5 | -5.4 | -5.7 |
| **BM (Init.)** | -4.4 | -5.0 | -4.3 | -4.6 |
| **BM (Reg.)** | **-1.3** | **-1.9** | **-1.8** | **-1.7** |

Table 3: The Impact of hidden dimension $d'_m$ on the accuracy of short-sequence continual learning benchmarks for short-sequences with T5-large.

| $d'_m$ | Order-1 | Order-2 | Order-3 | Avg |
|---|---|---|---|---|
| $d'_m=1$ | 64.9 | 66.4 | 64.9 | 65.4 |
| $d'_m=8$ | 73.3 | 73.5 | 68.1 | 71.6 |
| $d'_m=32$ | **74.9** | **74.0** | **74.1** | **74.3** |
| $d'_m=128$ | 74.3 | 74.3 | 70.7 | 73.1 |

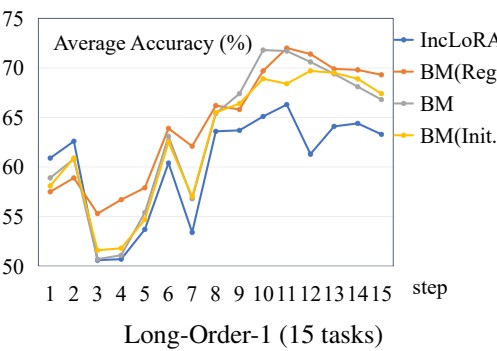

Long-Order-1 (15 tasks)

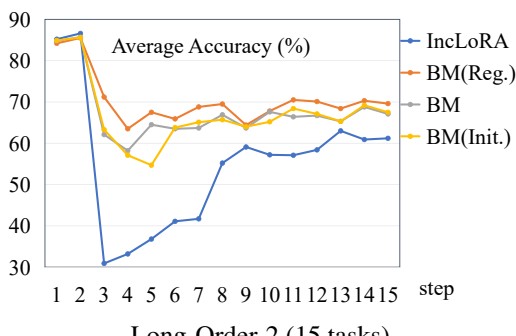

Long-Order-2 (15 tasks)

Figure 3: Learning trajectories on long-sequence continual learning benchmarks. Branching Memory demonstrates consistent stability and minimal forgetting across extended task sequences.

As shown in Figure 3, Branching Memory maintains stable performance with significantly reduced forgetting compared to baseline methods across both extended sequences. The regularization strategy proves particularly effective in preserving knowledge throughout the learning process. Quantitative results in Table 4 further validate these observations, demonstrating that Branching Memory with regularization achieves the best balance of high average accuracy and minimal backward transfer across both long sequences. These results confirm the method's strong scalability and effective forgetting mitigation in extended continual learning scenarios, with orthogonality regularization playing a crucial role in preserving previously acquired knowledge.

### 5.2.4 IMPACT OF HYPERPARAMETERS

**Hidden Dimension** $d'_m$: Table 3 shows the effect of varying the hidden dimension $d'_m$ of the lightweight memory branches on accuracy. Theoretically, increasing the hidden dimension can enhance the model's storage capacity, but experimental results indicate that an excessively large $d'_m$ can lead to increased computational overhead and decreased performance. The optimal hidden dimension is set to $d'_m = 32$, achieving the highest average accuracy on short-sequence tasks.

**Regularization Coefficient** $\lambda$: Table 5 shows the impact of the regularization parameter $\lambda$ on accuracy. An excessively large regularization coefficient can reduce the model's learning ability for new tasks, thereby affecting overall performance, while too small a coefficient may lead to increased forgetting of old tasks. In this experiment,

Table 4: Comprehensive performance on long-sequence benchmarks

| Method | Long-Order (15 tasks) | | Long-Order2 (15 tasks) | |
|---|---|---|---|---|
| | AA (%) | BWT (%) | AA (%) | BWT (%) |
| IncLoRA | 63.3 | -12.6 | 61.2 | -13.8 |
| **BM** | 66.8 | -5.4 | 67.1 | -5.3 |
| **BM (Init.)** | 67.4 | -5.3 | 67.5 | -4.7 |
| **BM (Reg.)** | **69.3** | **-2.3** | **69.6** | **-2.1** |

we set different regularization coefficients for different tasks, optimizing a suitable mixture, with specific parameters detailed in the appendix A.

Table 5: The impact of $\lambda$ on the accuracy of continual learning benchmarks for short-sequences with T5-large.

| $\lambda$ | Order-1 | Order-2 | Order-3 | Avg |
|---|---|---|---|---|
| $\lambda$=0.1 | 67.8 | 67.2 | 68.4 | 67.8 |
| $\lambda$=0.05 | 68.8 | 68.4 | 69.1 | 68.8 |
| $\lambda$=0.01 | 71.6 | 69.9 | 70.7 | 70.8 |
| $\lambda$=0.001 | 75.7 | 76.3 | 75.0 | 75.7 |
| $\lambda$=0.0005 | 77.0 | 76.5 | 75.1 | 76.2 |
| $\lambda$=0 | 75.4 | 73.1 | 74.1 | 74.2 |
| mixture | **77.3** | **77.1** | **75.5** | **76.6** |

Table 6: Performance on short-sequence continual learning benchmark with LLaMA-7B on short-sequence tasks. Comparison of IncLoRA and our methods

| Method | Order1 | Order2 | Order3 | Avg | BWT |
|---|---|---|---|---|---|
| IncLoRA | 75.4 | 75.2 | 75.1 | 75.2 | -3.8 |
| **BM** | 75.7 | 75.8 | 75.3 | 75.6 | -2.5 |
| **BM (Init.)** | 76.8 | 76.4 | 76.5 | 76.6 | -2.5 |
| **BM (Reg.)** | **77.1** | **76.5** | **77.2** | **76.9** | **-1.9** |

### 5.2.5 RESULTS ON OTHER LLMS

To evaluate the adaptability of our method on other LLMs, we tested it on Llama-7B. Table 6 shows that Branching Memory also improves the performance of this kind of model in continuous learning, while regularization and initialization strategies further enhance the model's memory ability. This indicates that our approach has good adaptability across different model architectures.

### 5.2.6 PARAMETER EFFICIENCY ANALYSIS

Table 7: Parameter efficiency comparison (T5-Large, 4 tasks)

| Method | Params/Task | Training Time (4 tasks) | GPU Memory | Inference Latency |
|---|---|---|---|---|
| IncLoRA (r=8) | 294,912 | 462.2s | 22GB | +0.3ms |
| BM ($d'_m$=8) | 98,304 | 349.1s | 21GB | +0.2ms |
| FullFT | 88M | 7637.4s | 22GB | +0ms |

Branching Memory demonstrates 2-3× better parameter efficiency than LoRA-based methods while maintaining competitive performance. The linear parameter growth (0.22% of T5-Large per task) remains practical even for 50-task sequences.

### 5.3 LIMITATIONS

The proposed Branching Memory method demonstrates strong continual learning capabilities but has several limitations. While our approach shows promising results on T5 and LLaMA2 architectures, its **scalability** to even larger models (*e.g.*, GPT-5 or DeepSeek) requires further validation. We experimented with extending branches to Multi-Head Self-Attention (MHSA) layers but observed only marginal performance gains (0.2%) with significant parameter overhead. Current experiments primarily focus on text classification tasks, and extending to more diverse task types such as question answering and mathematical reasoning would further validate generalizability. Additionally, while our inference design consolidates all branches into a unified model, more sophisticated branch selection mechanisms could potentially enhance performance in complex multi-task scenarios.

## 6 CONCLUSION

In this paper, we propose Branching Memory, a parameter-efficient continual learning framework based on task-specific lightweight branch expansions in transformer FFNs. Our approach effectively mitigates catastrophic forgetting in LLMs by isolating task-specific knowledge while preserving previously learned information through architectural expansion. The orthogonality-based regularization strategy significantly reduces task interference, while the consolidated inference design eliminates the need for complex branch selection mechanisms. Experimental results demonstrate that Branching Memory achieves superior parameter efficiency while maintaining competitive performance across diverse benchmarks. Validated on both T5 and LLaMA2 architectures with extended task sequences, our method provides a practical balance of performance, efficiency, and deployability for continual learning in dynamic NLP environments.

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

## A APPENDIX

### A.1 IMPLEMENTATION DETAILS

All experiments in this study were conducted on a server equipped with eight NVIDIA GeForce RTX 3090 GPUs, utilizing DeepSpeed and PyTorch frameworks for implementation. The training was performed with eight GPUs in parallel to improve efficiency. The main experimental parameters are detailed as follows:

**Number of epochs**: 1 epoch per task.

**Learning rate**: Constant at $1 \times 10^{-3}$.

**Batch size**: 64 in total (8 per GPU).

**Dropout rate**: 0.1.

**Weight decay**: 0.

**Extended Model Specifications:**

T5-Large: 770M parameters, hidden dimension $d = 1024$, FFN dimension $d_m = 4096$

LLaMA2-7B: 7B parameters, evaluated with the same branch dimensions

Branch dimensions tested: $d'_m \in \{1, 8, 16, 32, 128\}$

**Full Software Environment:**

```
transformers==4.28.0
datasets==1.17.0
deepspeed==0.10.0
fairscale==0.4.5
accelerate==0.20.3
nltk==3.8.1
tensorboard==2.13.0
tqdm==4.65.0
rouge_score==0.1.2
wandb==0.12.10
sentencepiece==0.1.96
protobuf==3.20.3
ninja==1.11.1
```

For the mixed regularization weight $\lambda$, the specific values vary across different task sequences. Refer to Table 8 for details.

**Initialization Parameters:**

Orthogonal initialization: $\sigma = 0.01$

Scaling factor: $\delta = 0.5$

Table 8: Regularization weight settings for different task sequences

| Task Sequence | Task Indices | $\lambda$ |
|---|---|---|
| Order 1-3 | 1, 2 | 0 |
| | 3, 4 | 0.001 |
| Long-Order 1-2 | 1, 2 | 0 |
| | 3+ | 0.0005 |

Initialization method: QR decomposition with Gaussian random matrix

**Initialization Procedure:** To improve the stability of training for new branches and reduce interference with previous tasks, we propose an initialization strategy Mishkin & Matas (2015) based on orthogonalization and distance control Farajtabar et al. (2020), which prevents excessive overlap between the parameters of new and previous branches.

Specifically, the parameters of the new branch for the $t$-th task, denoted as $W_{\text{task}}^{(t)}$, are initialized by first randomly sampling from a Gaussian distribution with mean 0 and variance $\sigma^2$. Then, QR decomposition Brock et al. (2021) is applied to generate an orthogonal matrix, which is further scaled by a diagonal matrix with hyperparameter $\delta_i$ to control the separation distance between branches. This ensures that the parameters of the new branch maintain orthogonality while being sufficiently distinct from previous branches.

**Parameter Selection:** The values $\sigma = 0.01$ and $\delta = 0.5$ were determined through empirical testing on validation sets, providing a balance between maintaining orthogonality and allowing sufficient learning capacity for new tasks.

## A.2 BASELINE DETAILS

We provide detailed descriptions of the baseline methods used for comparison in our experiments. These baselines represent a variety of approaches for continual learning and serve as references to evaluate the effectiveness of the proposed method.

**MTL:** Multi-task learning by jointly training on all tasks. This provides an upper-bound performance as the method has access to all tasks simultaneously during training.

**SeqFT de Masson D'Autume et al. (2019):** Standard sequential fine-tuning of all parameters on each task without any explicit mechanism to mitigate catastrophic forgetting.

**SeqLoRA**: A low-rank adaptation method that trains a single shared LoRA branch across all tasks. This approach does not differentiate task-specific adaptations, leading to limitations when handling task-specific knowledge.

**IncLoRA**: A low-rank adaptation method where each task is assigned its own LoRA branch. This allows task-specific representations to be learned, making it the primary comparison method in our experiments due to its similarity to the Branching Memory approach.

**Replay:** This method mitigates forgetting by replaying a subset of old task samples during new task training. It attempts to preserve knowledge from prior tasks using an explicit memory of past examples.

**EWC Kirkpatrick et al. (2017):** A regularization-based approach that penalizes updates to parameters that are deemed important for previous tasks. It uses a Fisher information matrix to estimate parameter importance.

**LFPT5 Qin & Joty (2021):** Continually trains a soft prompt that simultaneously learns to solve tasks and generate training samples. These generated samples are then used in experience replay to mitigate forgetting.

**L2P Wang et al. (2022b):** A method that uses the input to dynamically select and update prompts from a shared prompt pool in an instance-wise fashion. This approach enables efficient parameter reuse and task-specific adaptation.

**O-LoRA Wang et al. (2023a):** An orthogonal subspace-based incremental learning method. This method builds on LoRA by applying regularization constraints to the low-rank adapter parameters of new tasks, learning new tasks in an orthogonal subspace to preserve knowledge of old tasks.

**InfLoRA Wang et al. (2024):** A low-rank adaptation method that reparameterizes pre-trained weights and performs fine-tuning within carefully designed subspaces to eliminate interference between new and old tasks. It aims to achieve better trade-off between stability and plasticity by constraining the optimization space.

**GainLoRA Wang et al. (2025):** A gated integration approach for low-rank adaptation that expands new LoRA branches for each task while introducing gating modules to integrate new and old branches. The gating mechanism minimizes the influence from new LoRA branches on old tasks, effectively mitigating forgetting while maintaining plasticity for new task learning.

## A.3 DATASETS

This study utilized 15 datasets from the CL Benchmark Zhang et al. (2015), GLUE Wang et al. (2018), and SuperGLUE Wang et al. (2019), plus the IMDB movie review dataset Maas et al. (2011). These datasets cover various task types and domains. Details are provided in Table 9.

Table 9: Details of datasets used in the experiments

| Dataset | Task Type | Domain | Evaluation Metric |
|---------|-----------|--------|-------------------|
| Yelp | Sentiment Analysis | Yelp Reviews | Accuracy |
| Amazon | Sentiment Analysis | Amazon Reviews | Accuracy |
| DBpedia | Topic Classification | Wikipedia | Accuracy |
| Yahoo | Topic Classification | Yahoo Q&A | Accuracy |
| AG News | Topic Classification | News | Accuracy |
| MNLI | Natural Language Inference | Multiple | Accuracy |
| QQP | Paraphrase Detection | Quora | Accuracy |
| RTE | Natural Language Inference | News, Wikipedia | Accuracy |
| SST-2 | Sentiment Analysis | Movie Reviews | Accuracy |
| WiC | Word Sense Disambiguation | Vocabulary Database | Accuracy |
| CB | Natural Language Inference | Multiple | Accuracy |
| COPA | Question Answering | Blogs and Encyclopedias | Accuracy |
| BoolQ | Boolean Questions | Wikipedia | Accuracy |
| MultiRC | Question Answering | Multiple | Accuracy |
| IMDB | Sentiment Analysis | Movie Reviews | Accuracy |

## A.4 TASK SEQUENCE ORDERS

To comprehensively evaluate the continual learning ability, six task sequences are adopted. The first three follow the standard order from the CL Benchmark, while the last three are extended long-sequence tasks. Details are shown in Table 10.

Table 10: Task sequences used in the experiments

| Sequence | Order of Tasks |
|----------|----------------|
| Order-1 | DBpedia → Amazon → Yahoo → AG News |
| Order-2 | DBpedia → Amazon → AG News → Yahoo |
| Order-3 | Yahoo → Amazon → AG News → DBpedia |
| Long-Order-1 | MNLI → CB → WiC → COPA → QQP → BoolQA → RTE
→ IMDB → Yelp → Amazon → SST-2 → DBpedia → AG News → MultiRC → Yahoo |
| Long-Order-2 | RTE → CB → WiC → COPA → MNLI → QQP → BoolQA
→ IMDB → Yelp → SST-2 → MultiRC → AG News |

## A.5 TASK PROMPTS

To adapt to various tasks, specific prompts are designed for each task type, as listed in Table 11.

Table 11: Task prompts for different task types

| Task Type | Prompt |
|---|---|
| NLI | "What is the logical relationship between sentence 1 and sentence 2? Choose from the options." |
| QQP | "Do the first and second sentences have the same meaning? Choose from the options." |
| SC | "What is the sentiment of the following paragraph? Choose from the options." |
| TC | "What is the topic of the following paragraph? Choose from the options." |
| BoolQ | "Based on the given context, is the answer to the question true or false? Choose from the options." |
| MultiRC | "Based on the context and question, is the candidate answer correct or incorrect? Choose from the options." |
| WiC | "Given a word and two sentences, does the word have the same meaning in both sentences? Choose from the options." |

## A.6 STATEMENT ON LLM USAGE

In compliance with the ICLR 2026 policies on LLM usage, we disclose that LLMs were used solely for language refinement and improving the clarity of this paper. All research ideas, experimental design, data analysis, results, and conclusions were independently conducted and verified by the authors without any reliance on LLMs. The use of LLMs strictly adhered to the ICLR Code of Ethics, and the authors take full responsibility for the accuracy and integrity of the content presented in this work.

