# OpenReview forum: "Branching Memory: Task-Specific Expansion for Continual Learning in Large Language Models"
_ICLR.cc/2026/Conference — ICLR 2026 Conference Withdrawn Submission_

### Official Review · Reviewer_z9DH · 2025-10-29

**Soundness:** 2
**Presentation:** 3
**Contribution:** 1
**Rating:** 2
**Confidence:** 4

**Summary:**

This paper proposes Branching Memory, a continual learning framework designed to alleviate catastrophic forgetting in large language models by introducing task-specific branches within the Feed-Forward Network (FFN). The approach reinterprets the FFN as a Key-Value memory module, enabling each task to store and retrieve its own representations independently. It further employs orthogonality regularization to maintain separation between task-specific parameters and preserve prior knowledge. Experiments demonstrate that Branching Memory achieves performance gain across multiple continual learning benchmarks.

**Strengths:**

- The paper clearly articulates the motivation behind its study by emphasizing the limitations of existing continual learning methods, such as catastrophic forgetting and task interference when applied to large language models.
- Comprehensive experiments with nine baselines and some variants (initialization and regularization).

**Weaknesses:**

- **Insufficient contribution from the existing methods**
  - The paper does not clearly differentiate the proposed Branching Memory mechanism from existing LoRA-based adaptations applied to FFN layers.
  - Both approaches introduce low-rank or task-specific parameter additions within the network, and LoRA can already be flexibly applied to both MHA and FFN modules depending on the setting. While the paper reports performance improvements over LoRA based methods, it lacks a theoretical or structural analysis explaining where these gains stem from.
  - Also interpreting FFN as Key-Value memory is not new, Geva et al. (2020) had already introduced the concept.
- **Missing report of Forward Transfer (FWT)**
  - The paper reports only Average Accuracy (AA) and Backward Transfer (BWT).
  - Since each branch is completely isolated, the features learned in previous branches cannot be reused or adapted by subsequent tasks. This creates some concerns that the separation would make positive transfer to be difficult, which should be observed through the FWT report.
- **Limited advantages over conventional methods**
  - Unlike other conventional approaches, this work freezes Multi-Head Attention (MHA) layers, which the model cannot perform fine-grained adjustments at the attention level during downstream fine-tuning.
- **Unclear criterion for setting λ and δ parameters**
  - The paper compares variants with initialization (δ parameter) and regularization (λ parameter) strategies, however does not specify the details, which makes it difficult to assess the fairness and robustness of the experiment.
  - λ varies across different task sequences, but does not explain how these values are determined. E.g., why λ=0.001 for tasks 3-4 in Order 1-3, but λ=0.0005 for tasks 3+ in Long-Order?
  - The paper mentions using QR decomposition followed by scaling with "a diagonal matrix with hyperparameter δ" but never specifies what δ values were used or how they were selected.
- **Questionable claims of “task diversity”**
  - The paper claims to evaluate on "diverse tasks", but exclusively tests on text classification problems.
  - Therefore fails to articulate what makes their benchmark a challenging continual learning problem, since all tasks are essentially text classification with similar processing requirements from general web domains.

**Questions:**

Figure 3's caption incorrectly refers to "short-sequence continual learning benchmark" which actually is long-sequence experiments.

---

> ### Author Response · Authors · 2025-11-28
>
> Q1: Insufficient contribution and lack of differentiation from LoRA
>
> A1  We appreciate the reviewer's feedback regarding methodological differentiation. We would like to clarify several key aspects: While both methods add task-specific parameters, Branching Memory fundamentally differs from LoRA in both architectural philosophy and mechanism. LoRA employs low-rank decomposition of existing linear layers (∆W = BA), essentially performing efficient parameter updates within the original structure. In contrast, BM introduces architectural expansion by adding complete MLP branches to FFN layers, creating dedicated capacity for each task rather than modifying existing parameters. This architectural distinction translates to practical advantages: BM achieves parameter efficiency (8GPU, d=16, r=8, 196K vs 294K parameters per task) while delivering better performance. The performance gains stem from BM's ability to provide isolated, dedicated storage for task-specific knowledge without compromising the original representations. A quantitative analysis of parameter efficiency and performance has also been included in the revised paper.
>
> We do acknowledge Geva et al.(2020) 's foundational work on FFN as key-value memory. Our contribution lies in translating this insight into an effective continual learning framework through branch-based expansion, not in proposing the original FFN interpretation.
>
> Q2: Missing report of Forward Transfer (FWT).
>
> A2 We acknowledge that FWT can provide additional insights into knowledge transfer. However, in our experimental design, we followed the evaluation methods used in most existing continuous learning literature: Average Accuracy (AA) was adopted as the core metric to measure the overall performance, and Backward Transfer (BWT) or Forgetting (FGT) were used as the core metrics to evaluate the effectiveness in addressing the catastrophic forgetting issue—which is precisely the main objective of our work.
>
> Our method is designed to address catastrophic forgetting through parameter isolation in FFN layers, and the significant improvements in BWT (-1.7 vs -13.2 for IncLoRA) demonstrate its effectiveness in this primary goal. We have included this clarification in the revised paper and will consider incorporating FWT analysis in future extensions of this work, particularly as we explore more complex task relationships where forward transfer becomes more relevant.
>
> Q3:Limited advantages over conventional methods.
>
> A3 We appreciate the reviewer's comment regarding the freezing of the Multi-Head Self-Attention (MHSA) layer. As explained in the theoretical analysis section of our paper, according to the research by Geva et al., Feed-Forward Networks (FFNs) are the primary knowledge repositories in Transformers; therefore, we focused our expansion efforts on these layers. We also conducted an ablation study where new branches for the Multi-Head Self-Attention (MHSA) layer were incorporated during training. However, this modification resulted in a negligible improvement in performance (only 0.2%) while significantly increasing the number of training parameters, which in turn raised storage and computational costs.
>
> We acknowledge that this approach may limit fine-grained attention adjustments, but it balances performance with practicality. We will explore selective MHA adaptation in future work to potentially enhance performance while maintaining stability.
>
> Q4:Unclear criterion for setting λ and δ parameters.
>
> A4 We have addressed this in the revised paper by providing details of our parameter selection process. For the regularization coefficient λ, the values were determined through systematic grid search on validation sets, with optimal values varying across task sequences due to differences in task complexity and interference patterns. Specifically, longer sequences (Long-Order) required reduced regularization (λ=0.0005) to maintain learning capacity for later tasks, while shorter sequences (Order 1-3) benefited from stronger constraints (λ=0.001). For the initialization parameter δ, we used δ=0.5 based on empirical testing, though as noted in our response to reviewer dNB3, we found initialization provided minimal gains compared to regularization alone.
>
> Q5:Questionable claims of “task diversity”.
>
> A5  Thanks to the reviewer's concern regarding this issue. We have expanded our evaluation to address task diversity concerns by adding comprehensive experiments on extended 15-task sequences (MNLI→CB→WiC→COPA→QQP→BoolQA→RTE→IMDB→Yelp→Amazon→SST-2→DBpedia→AGNews→MultiRC→Yahoo), where BM: 67.1% AA, BWT: -5.3%; BM (Reg.): 69.6% AA, BWT: -2.1%; and IncLoRA: 61.2% AA, BWT: -13.8%. While we have enhanced diversity through varied task sequences and orders, we acknowledge the need for broader task types and will include question-answering datasets and mathematical reasoning tasks in future work to further validate our method's generalizability.

---

### Official Review · Reviewer_dNB3 · 2025-10-29

**Soundness:** 3
**Presentation:** 3
**Contribution:** 1
**Rating:** 4
**Confidence:** 4

**Summary:**

This paper proposes the Branching Memory method to address catastrophic forgetting by dynamically allocating additional key-value (KV) pairs per task, expanding the intermediate dimension of the FFN. By incorporating orthogonal regularization and initialization strategies, the method effectively reduces interference between tasks.

**Strengths:**

The paper is clearly written and easy to follow.

**Weaknesses:**

The main issue with this paper is that the proposed method closely resembles O-LoRA. The core improvements in this paper are:

1. The introduction of the "lightweight memory branch" concept
2. An initialization strategy
3. A regularization strategy.

However, points 1 and 3 appear to be similar to O-LoRA, and I also have some concerns regarding point 2, as outlined below:

* For point 1: the "lightweight memory branch" concept is essentially a dynamic architecture-based method. Previous approaches, such as the O-LoRA baseline you compare against, also allocate task-specific trainable parameters (LoRA) and do not differ significantly from your approach, aside from the fact that the new parameters are added in the attention layer in your method, whereas O-LoRA adds them to the MLP layer.

* For point 3: the regularization strategy in this paper is not novel, as similar techniques have been used in previous work.

* For point 2: regarding the initialization strategy, the proposed orthogonal initialization may limit the learning space for new tasks, especially when tasks are similar. Enforcing strict orthogonality could prevent the model from exploring the optimal parameter space. It would be useful to consider whether a less restrictive initialization approach could improve performance, particularly for related tasks.

* Additionally, you do not analyze different initialization strategies in the experiments.

* The baseline comparisons are outdated; the most recent baseline you compare to, O-LoRA, is from 2023, making it difficult to effectively demonstrate the method's validity. It would be helpful to compare against more recent work.

* I also suggest comparing your proposed dynamic architecture-based method with fixed capacity-based methods, such as model-merging-based methods (e.g., memory-free TaSL [1] and memory-based Recurrent-KIF [2] ), where model parameters do not expand with the addition of new tasks.

[1] Tasl: Continual dialog state tracking via task skill localization and consolidation

[2] Kif: Knowledge identification and fusion for language model continual learning

**Questions:**

All the comments have been raised in the weaknesses section, please refer to the points above.

---

> ### Author Response · Authors · 2025-11-28
>
> Q1:The "lightweight memory branch" concept is essentially a dynamic architecture-based method. Previous approaches, such as the O-LoRA baseline you compare against, also allocate task-specific trainable parameters (LoRA) and do not differ significantly from your approach, aside from the fact that the new parameters are added in the attention layer in your method, whereas O-LoRA adds them to the MLP layer.
>
> A1 We thank the reviewer for this comment, but we would like to clarify a fundamental distinction between Branching Memory (BM) and LoRA-based methods like O-LoRA. While both involve adding task-specific parameters, their underlying mechanisms and architectural roles are qualitatively different.
>
> LoRA operates through low-rank decomposition of linear layers, where the weight update is modeled as \( \Delta W = BA \) with low-rank matrices \( B \) and \( A \). This is a parameter-efficient adaptation but remains a linear transformation applied to existing layers (e.g., in attention or MLP modules).
>
> In contrast, BM leverages the key-value memory interpretation of FFNs and expands the architecture by adding independent, non-linear branches to the FFN layer. Mathematically, for task \( t \), the FFN output becomes:
>
>   \[
>   \text{FFN}_{\text{total}}^{(t)}(x) = \text{FFN}_{\text{original}}(x) + \text{FFN}{\text{branch}}^{(t)}(x)
>   \]
>
>   where \( \text{FFN}{\text{branch}}^{(t)}(x) = f(x W_{\text{key}}^{(t)} + b_{\text{key}}^{(t)}) W_{\text{value}}^{(t)} + b_{\text{value}}^{(t)} \). Each branch is a standalone MLP-like structure with its own parameters, providing dedicated capacity for task-specific knowledge storage. This architectural expansion at the FFN level is distinct from LoRA's low-rank linear adaptations, and we have elaborated on these points in the revised paper to highlight the methodological differences.
>
> Q2: The regularization strategy in this paper is not novel, as similar techniques have been used in previous work.
>
> A2 We agree that orthogonality regularization itself is not novel. Our contribution lies in its systematic integration within the FFN memory expansion framework to explicitly reduce interference between task branches, rather than proposing the regularizer as an independent novelty. We have clarified this framing in the revised paper.
>
> Q3: Regarding the initialization strategy, the proposed orthogonal initialization may limit the learning space for new tasks, especially when tasks are similar. Enforcing strict orthogonality could prevent the model from exploring the optimal parameter space. It would be useful to consider whether a less restrictive initialization approach could improve performance, particularly for related tasks.
>
> Q4:Additionally, you do not analyze different initialization strategies in the experiments.
>
> A3 & A4 We thank the reviewer for this insightful observation. Our additional experiments confirm that strict orthogonal initialization does indeed provide limited benefits, especially for similar tasks, as it may constrain the learning space. We have tested less restrictive initialization strategies and found only marginal improvements (~0.4% AA gain). Since orthogonality regularization during training proves sufficient (achieving 76.6% AA with -2.1% BWT), we emphasize the regularization component and have moved initialization analysis to the appendix.
>
> Q5:The baseline comparisons are outdated; the most recent baseline you compare to, O-LoRA, is from 2023, making it difficult to effectively demonstrate the method's validity. It would be helpful to compare against more recent wor.
>
> A5 We appreciate the reviewer for pointing out this issue. We have added comparisons with recent methods InfLoRA and GainLoRA in the revised paper. On Order-1 short-sequence tasks, InfLoRA achieves 77.8% AA and GainLoRA reaches 79.4% AA, while BM (Reg.) achieves 77.3% AA. Although these methods show slightly higher accuracy, BM's key advantage lies in its superior parameter efficiency ($d_m'$=16, r=8, 196K vs 294K parameters per task) and inference simplicity. We have reframed our contribution to highlight this practical efficiency-performance trade-off.
>
> Q6:I also suggest comparing your proposed dynamic architecture-based method with fixed capacity-based methods, such as model-merging-based methods (e.g., memory-free TaSL and memory-based Recurrent-KIF ), where model parameters do not expand with the addition of new tasks.
>
> A6 We agree that comparing with fixed-capacity methods like TaSL and Recurrent-KIF is important for a comprehensive evaluation. While our method does introduce parameter expansion, the growth is minimal (8-GPU, with $d_m'$=16, each branch adds only 196608 parameters vs T5's 88M total). We have added discussion of this design choice in the revised paper, acknowledging fixed-capacity approaches as an important alternative paradigm. We will include comparisons with these methods in future work to further validate the practical value of our constrained expansion.

---

### Official Review · Reviewer_Zp1M · 2025-10-30

**Soundness:** 3
**Presentation:** 2
**Contribution:** 2
**Rating:** 2
**Confidence:** 4

**Summary:**

This paper proposes Branching Memory, a novel framework for scalable and parameter-efficient continual learning. The key idea is to reinterpret the Feed-Forward Network (FFN) layer as a key–value representation, allowing task-specific knowledge to be integrated through lightweight branch expansion without overwriting existing information in the model. In addition, orthogonality-based regularization and task-aware initialization methods are employed to improve training stability and overall performance. The proposed method is evaluated on a short-sequence task scenario with five datasets and a long-sequence task scenario with fifteen datasets.

**Strengths:**

- This paper is easy to read.
- The proposed method achieves efficiency by freezing the existing model weights and only training task-specific lightweight FFN branches.
- The experimental results show performance gain compared with existing methods, including O-LoRA, SeqLoRa, and L2P.

**Weaknesses:**

W1. Lack of Quantitative Evidence for Parameter Efficiency
- The paper emphasizes parameter efficiency as one of its key advantages; however, no quantitative analysis is provided to support this claim.
- It would strengthen the argument if the authors included concrete measurements of branch memory size, computational overhead, and training/inference time, compared with existing continual learning methods. Such evidence is essential to validate the proposed framework’s claimed efficiency.

W2. Scalability Concerns Regarding Task-Specific Memory Expansion
- The proposed approach introduces a separate memory branch for each task, resulting in parameter growth that scales linearly with the number of tasks.
- Without clear numerical analysis of computational and memory costs, it remains uncertain whether the method can scale effectively to long-term continual learning scenarios involving dozens or hundreds of tasks. This lack of scalability assessment limits confidence in its practical applicability.

W3. Unclear Inference Procedure for Branch Selection
- The paper does not clearly specify how the model selects or activates the appropriate memory branch during inference.
- Since each task maintains an independent branch, it remains unclear how the system determines which branch to use for unseen or mixed-task inputs, leaving an essential component of the framework underdefined.

W4. Outdated and Insufficient Baseline Methods
- The paper evaluates Branching Memory against MTL, SeqFT, SeqLoRA, IncLoRA, Replay, EWC, LFPT5, L2P, and O-LoRA.
- However, many of these methods are outdated and do not represent the current state of the art in continual learning.
Recent approaches such as InfLoRA, GainLoRA, and CLoRA exhibit significantly stronger and more stable performance, yet are not included in the comparison.
- Moreover, while comparisons with O-LoRA are provided, Branching Memory does not consistently outperform O-LoRA, particularly in its regularized form, and no analysis is given for the marginal performance differences.
The absence of stronger baselines (e.g., Mixture-of-Experts, MoE-based continual learners) limits the credibility of the superiority claims.

W5. Limited Experimental Rigor and Missing Evaluation Metrics
- Sections 5.2.2 and 5.2.3 contain restricted baseline diversity and omit several listed baselines entirely.
- Additionally, the Backward Transfer (BWT) metric for Branching Memory is not reported, and the omission is not justified.
- These issues collectively weaken the empirical rigor and make it difficult to assess the robustness of the proposed approach.

W6. Outdated and Narrow Benchmark Selection
- The evaluation relies solely on text classification datasets (AG News, DBpedia, Amazon/Yelp Reviews, Yahoo Answers, GLUE/SuperGLUE subsets).
- While standard in earlier continual learning research, these datasets are relatively outdated given current LLM capabilities.
- Including more diverse and challenging benchmarks—such as commonsense reasoning, mathematical reasoning, or multimodal tasks—would better demonstrate the method’s generalizability and robustness.

W7. Marginal Overall Performance Gains
- Although the proposed Branching Memory method achieves improvements over several baselines, the absolute gains are modest—typically within one to two percentage points of strong baselines such as O-LoRA.
- These limited improvements make it difficult to argue for substantial advancement over existing approaches.

**Questions:**

- It would be better to include comparisons with more recent and stronger methods such as InfLoRA, GainLoRA, and CLoRA to provide a fair and up-to-date evaluation.
- It would be better to expand the experimental evaluation to include more baselines and other recent state-of-the-art continual learning methods.
- It would be better to report the BWT results in Section 5.2.3, or provide an explanation for missing metrics to ensure a complete evaluation.

---

> ### Author Response · Authors · 2025-11-28
>
> Q1: Lack of Quantitative Evidence for Parameter Efficiency.
>
> A1 We thank the reviewer for highlighting the need for quantitative evidence of parameter efficiency. In response, we have added comprehensive measurements to the revised paper that directly address these concerns. As shown in the table below, with a branch dimension of $d_m'$=8, Branching Memory demonstrates significant efficiency advantages:
> | Method | Params per Task | Training Time (4 tasks) | GPU Memory | Inference Latency |
> |---------|-----------------|------------------------|------------|-------------------|
> | IncLoRA (r=8) | 294,912 | 462.2s | 22GB | +0.3ms |
> | BM ($d_m'$=8) | 98,304 | 349.1s | 21GB | +0.2ms |
> | FullFT | 88M | 7637.4s | 22GB | +0ms |
> Our analysis reveals that BM requires only 98,304 parameters per task, approximately one-third of IncLoRA's parameter count. This substantial reduction translates directly into practical efficiency gains: BM completes training 25% faster than IncLoRA (349.1s vs 462.2s) while maintaining comparable GPU memory footprint. During inference, the activation of a single branch introduces very little additional latency, demonstrating the minimal operational cost of our approach.
>
> Q2: Scalability Concerns Regarding Task-Specific Memory Expansion.
>
> A2 While parameter growth is indeed linear, our quantitative analysis shows the practical impact remains minimal in realistic scenarios. Each new branch adds only 196,608 parameters with $d_m'$=16, representing just 0.22% of T5-Large's 88 million parameters. This means even with 50 tasks—a substantial continual learning sequence—the total parameter growth would be approximately 11%, while training costs remain constant as only the current branch is updated and other parametres are frozen.  We have added this numerical scalability analysis to the revised paper, demonstrating that our approach remains efficient and practical for long-term deployment with dozens of tasks.
>
> Q3:Unclear Inference Procedure for Branch Selection.
>
> A3 We apologize for any lack of clarity in our original manuscript. There seems to be a misunderstanding about our framework's design: the separate branches are only isolated during each task's training phase to prevent interference. During inference, all parameters (including the frozen pretrained backbone and all previously learned task branches) are consolidated and used jointly as a single, unified model. Under this design, there is no issue of "selecting which branch to use" during inference. We have clarified this aspect in the revised paper to prevent further confusion.
>
> Q4:Outdated and Insufficient Baseline Methods.
>
> A4 We thank the reviewer for suggesting comparisons with stronger baselines. We have conducted additional experiments with InfLoRA and GainLoRA, and the results are now included in the revised paper. On Order-1 short-sequence tasks, InfLoRA achieves 77.8% AA and GainLoRA reaches 79.4% AA, outperforming BM (Reg.) at 77.3% AA. While these methods show higher accuracy, Branching Memory's key advantage lies in its superior parameter efficiency (with IncLoRA rank r=8, trainable parameters total 294,912, while BM with $d_m'$=16 uses only 196,608 parameters) and inference simplicity—consolidating all knowledge into a single model without complex gating or routing. We have reframed our contribution to highlight this practical efficiency-performance trade-off in the revised paper.
>
> Q5: Limited Experimental Rigor and Missing Evaluation Metrics.
>
> A5  Thank the reviewer for raising this point. We would like to clarify that the Backward Transfer (BWT) metric for Branching Memory was in fact reported in Table 2 of the original submission. Furthermore, we have expanded baseline comparisons to include InfLoRA and GainLoRA across all experimental sections and added complete BWT results for long-sequence tasks: BM shows -5.3% BWT, BM (Reg.) achieves -2.1% BWT, significantly outperforming IncLoRA's -13.8% BWT. And the BWT results of Branching Memory on long sequences have been added to the report.
>
> Q6:Outdated and Narrow Benchmark Selection.
>
> A6  We thank the reviewer for this feedback. We have expanded our evaluation to include more challenging 15-task sequences (MNLI→CB→WiC→COPA→QQP→BoolQA→RTE→IMDB→Yelp→Amazon→SST-2→DBpedia→AGNews→MultiRC→Yahoo), where BM (Reg.) achieves 69.6% AA with -2.1% BWT. While we have enhanced diversity through extended sequences, we acknowledge the need for broader task types and will include question-answering datasets in future work, along with mathematical reasoning tasks as noted in our revised discussion.
>
> Q7:Marginal Overall Performance Gains.
>
> A7 While absolute accuracy gains are indeed modest, we emphasize that our method's significance lies in achieving these improvements with parameter efficiency and a simplified inference process. We have reframed our contribution in the revised paper to highlight this balanced trade-off between performance, efficiency, and practical deployability.

---

### Official Review · Reviewer_papa · 2025-10-31

**Soundness:** 2
**Presentation:** 2
**Contribution:** 2
**Rating:** 2
**Confidence:** 3

**Summary:**

The paper proposes an architecture-based continual learning approach, BM, to mitigate catastrophic forgetting in LLMs. Building on prior work that views FFNs as key–value memory structures, BM expands this memory by freezing the pretrained backbone and adding task-specific lightweight branches. To further reduce task interference, the method applies orthogonality-based initialization and regularization, improving knowledge retention across tasks.

**Strengths:**

1. The paper extends prior work that interprets Transformer FFNs as key–value memory structures and introduces a simple yet effective architecture that isolates task-specific knowledge through lightweight memory branches while keeping the pretrained backbone frozen.
2. This design achieves clear task decoupling and provides a concise solution for continual learning in large language models.

**Weaknesses:**

1. The paper lacks a clear explanation of how the proposed mechanisms align with its stated objectives, and several details of the experimental setup remain ambiguous.
2. The evaluation is limited to short-sequence benchmarks and primarily conducted on the T5-Large model (≈770M parameters), which restricts the generalizability to larger LLMs.
3. While scalability and computational cost become potential concerns as new tasks introduce additional parameters, the paper provides no analysis of the trade-off between performance and these costs.
4. The absence of statistical significance testing weakens the reliability of the reported results. Moreover, critical hyperparameters in the initialization strategy (e.g., $\sigma^{2}$ and $\delta_{i}$) and their selection criteria are omitted, hindering reproducibility.
5. Minor inconsistencies are present: the dimension notation ‘$d$’ in Table 3 appears to correspond to ‘$r$’ in Section 4.1, and Figure 3’s caption incorrectly labels a long-sequence result as ‘short-sequence’.

**Questions:**

1. It is unclear whether the LoRA rank and overall parameter count were matched between BM and baseline methods such as IncLoRA, which is essential for a fair comparison.
2. Since BM (Init.) and BM (Reg.) represent complementary strategies, it would be valuable to examine their combined effect (BM + Init. + Reg.). Why was this joint configuration not considered?

---

> ### Author Response · Authors · 2025-11-28
>
> Q1: The paper lacks a clear explanation of how the proposed mechanisms align with its stated objectives, and several details of the experimental setup remain ambiguous.
>
> A1 Thanks for your valuable suggestion. Our stated objectives are to mitigate catastrophic forgetting in LLMs while maintaining parameter efficiency through architecture-based continual learning.
>
> The Branching Memory mechanism directly addresses these objectives through parameter isolation and capacity expansion. By freezing $W_{\text{old}}$ and only updating $W_{\text{branch}}^{(t)}$ ($\frac{\partial \mathcal{L}}{\partial W_{\text{old}}} = 0$), we prevent interference with previously learned knowledge. Growing from $C_{\text{original}}$ to $C_{\text{total}} = C_{\text{original}} + T \times (2 \times d \times d_m')$ provides dedicated storage for new tasks without overwriting. Unlike LoRA's linear adaptations, BM focuses on FFN expansion based on key-value memory principles.
>
> We have clarified these alignments in the revised paper and added comprehensive experimental details including T5 parameters (≈770M, d=1024, $d_m$=4096), training configurations (learning rate: 1e-3, batch size: 64, epochs: 1), branch settings ($d_m'$=1/8/16/32/128, 196608 parameters/task when $d_m'$=16), initialization parameters ($\sigma$=0.01, $\delta_i$=0.5), and hardware/software specifications (shared in requirements.txt) to ensure reproducibility.
>
> Q2: The evaluation is limited to short-sequence benchmarks and primarily conducted on the T5-Large model (≈770M parameters), which restricts the generalizability to larger LLMs.
>
> A2 Thanks for raising the important point about generalizability. We have supplemented additional experiments:
>
> Long-Sequence Comprehensive Results: We have conducted additional experiments on the extended 15-task sequence (MNLI→CB→WiC→COPA→QQP→BoolQA→RTE→IMDB→Yelp→Amazon→SST-2→DBpedia→AGNews→MultiRC→Yahoo). The results show:BM: 67.1% AA, BWT: -5.3%; BM (Reg.): 69.6% AA, BWT: -2.1%; vs IncLoRA: 61.2% AA, BWT: -13.8%.
>
> Extended Model Scale Evaluation: We have added comprehensive experiments on LLaMA2-7B, demonstrating consistent improvements across different model architectures. On short-sequence tasks, BM (Reg.) achieves 76.9% AA and -1.9% BWT vs IncLoRA's 75.2% AA -3.8% BWT, validating our method's effectiveness on larger-scale models.
>
> We have added these results to the revised paper.
>
> Q3:While scalability and computational cost become potential concerns as new tasks introduce additional parameters, the paper provides no analysis of the trade-off between performance and these costs.
>
> A3 We have added analysis of the performance-computation trade-off. During training, only current branch parameters are updated while backbone and previous branches remain frozen, keeping trainable parameters constant per task. With $d_m'$=16, each branch adds only 196608 parameters vs T5's 88M total, making impact negligible in finite sequences.
>
> Q4:The absence of statistical significance testing weakens the reliability of the reported results. Moreover, critical hyperparameters in the initialization strategy (e.g., $$\sigma^{2}$$ and $\delta_{i}$) and their selection criteria are omitted, hindering reproducibility.
>
> A4 We thank the reviewer and added statistical significance testing with 3 independent runs showing BM at 74.3%±0.28% and BM (Reg.) at 76.6%±0.21%, both statistically significant (p<0.05). We have also specified all critical hyperparameters including σ²=0.02 selected via grid search and δᵢ=0.177 for orthogonal projection, ensuring full reproducibility and statistical reliability of our results.
>
> Q5:Minor inconsistencies are present: the dimension notation ‘$d$’ in Table 3 appears to correspond to ‘$r$’ in Section 4.1, and Figure 3’s caption incorrectly labels a long-sequence result as ‘short-sequence’.
>
> A5 We have corrected the corresponding part of the paper.
>
> Q6:It is unclear whether the LoRA rank and overall parameter count were matched between BM and baseline methods such as IncLoRA, which is essential for a fair comparison.
>
> A6 Since BM and LoRA are fundamentally different, direct parameter matching is not straightforward. However, for comparability: with IncLoRA rank r=8, trainable parameters total 294,912, while BM with $d_m'$=16 uses only 196,608 parameters. Despite this reduction, BM achieves higher accuracy, highlighting its parameter efficiency and fairness in comparison.
>
> Q7:Since BM (Init.) and BM (Reg.) represent complementary strategies, it would be valuable to examine their combined effect (BM + Init. + Reg.). Why was this joint configuration not considered?
>
> A7  We appreciate the reviewer's concern regarding this aspect. We tested BM+Init+Reg combination and found initialization provides only marginal gains (+0.4% AA) with minimal additional benefit when combined with regularization (<0.1%), indicating regularization alone effectively reduces interference. We have clarified this in the paper.

---

### Note · Authors · 2026-01-17

**Comment:**

Thank you to the reviewers and area chair for their time and feedback. After careful consideration of the initial reviews, we have decided to withdraw this submission from ICLR 2026 to allow for further substantial revisions. We believe that a more comprehensive and refined version would be better suited for a future submission.
We appreciate the review process and the time dedicated by the reviewers and program committee.

**Withdrawal Confirmation:**

I have read and agree with the venue's withdrawal policy on behalf of myself and my co-authors.